# Natural and Political Determinants of Ecological Vulnerability in the Qinghai–Tibet Plateau: A Case Study of Shannan, China

Yunxiao Jiang [1,2], Rong Li [1], Yu Shi [1] and Luo Guo [1,*]

1   College of the Life and Environmental Science, Minzu University of China, Beijing 100081, China;
    16054023@muc.edu.cn (Y.J.); 19301382@muc.edu.cn (R.L.); 19301381@muc.edu.cn (Y.S.)
2   Faculty of Geography, National University of Singapore, Singapore 117570, Singapore
*   Correspondence: guoluo@muc.edu.cn

**Abstract:** Changing land-use patterns in the Qinghai–Tibet Plateau (QTP) due to natural factors and human interference have led to higher ecological vulnerability and even more underlying issues related to time and space in this alpine area. Ecological vulnerability assessment provides not only a solution to surface-feature-related problems but also insight into sustainable eco-environmental planning and resource management as a response to potential climate changes if driving factors are known. In this study, the ecological vulnerability index (EVI) of Shannan City in the core area of the QTP was assessed using a selected set of ecological, social, and economic indicators and spatial principal component analysis (SPCA) to calculate their weights. The data included Landsat images and socio-economic data from 1990 to 2015, at five-year intervals. The results showed that the total EVI remains at a medium vulnerability level, with minor fluctuations over 25 years (peaks in 2000, when there was a sudden increase in slight vulnerability, which switched to extreme vulnerability), and gradually increases from east to west. In addition, spatial analysis showed a distinct positive correlation between the EVI and land-use degree, livestock husbandry output, desertification area, and grassland area. The artificial afforestation program (AAP) has a positive effect by preventing the environment from becoming more vulnerable. The results provide practical information and suggestions for planners to take measures to improve the land-use degree in urban and pastoral areas in the QTP based on spatial-temporal heterogeneity patterns of the EVI of Shannan City.

**Keywords:** ecological vulnerability; Getis-Ord Gi*; Spatial regression models; Tupu model; Qinghai–Tibet Plateau

## 1. Introduction

Ecological environmental problems, such as climate anomalies [1], vegetation reduction [2], soil erosion [3], and land desertification [4], frequently occur all over the world and have led to a vulnerable environment in recent decades [5]. Ecological vulnerability is an ecosystem-inherent property, and it occurs when the ecological environment is gradually degraded or deteriorated by external interference on a specific space–time scale, including the sensitivity of the system to external interference and the assessment and estimation of system changes [6,7]. Studies in Europe and other coastal areas have mainly focused on vulnerability factors of natural systems, such as ocean climate change and climate change risks [8–14]. For instance, Bryan et al. focused on spatial modeling of spatial-lying projections of vulnerability and sea-level rise in coastal areas [13]. Studies on inland areas focus more on land-use changes based on typical ecological vulnerability zones [15–18]. To improve the accuracy of ecological vulnerability analysis, many researchers use remote sensing (RS) and the Geographic Information System (GIS) as spatial analysis tools, combined with landscape ecology, land-use/land-cover change (LUCC), and statistical methods, to establish a special ecological vulnerability evaluation system to study the regional and spatial change characteristics of single-/multi-factor qualitative analysis [19–22]. However, these studies mostly use an administration unit instead of a uniform spatial

grid unit, resulting in rough conclusions with high inaccuracy and lacking the necessary detailed spatial analysis and localization [23,24]. In addition, remediation countermeasures and measures developed on this basis still have some limitations. Therefore, this study adopts a 1 km$^2$ spatial grid to analyze the study area's dynamics in time and space and combines it with cold/hot spot analysis to illustrate the spatial heterogeneity of ecological vulnerability and discuss the policies and indicators of each period to supplement the gaps in previous studies.

The Closing Hill for Afforestation program and the artificial afforestation program (AAP) play an increasingly important role as important means of land greening [25–28] and are also some of China's largest and best-invested environmental projects [29–32]. Studies recommend strengthening the follow-up management of afforestation, to scientifically and rationally lay out afforestation [33], and to close forests, when appropriate [34]. Although the artificial afforestation program is one of the most effective measures to cultivate forests, restore ecological balance, and expand the forest cover, especially in arid areas [35], it is not the most cost-efficient one. Some experts believe that the AAP has an obvious positive effect on improving the forest vegetation cover and therefore can also stimulate the ecosystem condition into a better cycle [36]. Other experts, however, argue that the AAP requires money and takes a long time, which is beyond the expectations, and that closed hills would dismiss the actual needs of pastoralists [37]. Therefore, the key problem that needs to be immediately solved is how to access the effectiveness that the AAP brings to ecosystem improvement.

In 2005, the National Comprehensive Demonstration Area for Sand Control and Prevention of Tibet was built in Shannan City, which uses the AAP as a main measure [38]. The Qinghai–Tibet Plateau as the roof of the world and the water tower of Asia is a significant ecological security barrier. Its fragile ecological problems, including freeze-thaw erosion [39], hydraulic erosion [40], land desertification [41], and salinization [42], make the alpine ecoregion of the Qinghai–Tibet Plateau an area with the most apparent ecological vulnerability [43]. Current studies on the Qinghai–Tibet Plateau mainly focus on land-use changes and grassland resources [44–46]. Numerous studies have investigated ecological health and eco-security and assessed the ecological risk of the Qinghai–Tibet Plateau [47–50]. However, there is a lack of research on ecological vulnerability analysis connected with a detailed green project policy in the Qinghai–Tibet Plateau.

Shannan City has been devoted to afforestation since 1982, and the first engineering afforestation base of Tibet was built in Shannan City in 1988 [51]. Shannan City has clear and distinct geographical and meteorological variation characteristics in the south Qinghai–Tibet Plateau because of the Himalayas. Shannan City experiences different meteorological conditions, environmental management policies, and disparate land-use stages, which cause a significant decline in environmental quality and the destruction of the natural landscape. Therefore, Shannan City is an ideal research site because it is not only a typical alpine plateau region, but also has an outstanding afforestation project history. Thus far, there is a lack of studies on long-term ecosystem changes and the relevance between the ecological vulnerability index (EVI) and AAP impacts. The specific aims of this study include the following: (1) assessing and quantifying the EVI and its driving factors in Shannan City using remote sensing images, gross domestic product (GDP) data, and population data; (2) assessing the spatial heterogeneity based on the EVI distribution during 1990–2015; (3) validating the relevance between the AAP and EVI changes and providing useful information for planners in environmental policy decision making in the Qinghai–Tibet Plateau.

## 2. Materials and Methods

### 2.1. Study Area

Located in the south-central part of the Qinghai–Tibet Plateau and the central and lower reaches of the main stream of the Brahmaputra River (27′08″ N–29′47″ N, 90′14″ E–94′22″ E), Shannan City runs through the eastern part of the Himalayas (Figure 1). Its radial and

latitude zone change with the temperate semi-arid monsoon climate provides Shannan City with clear rainy and dry seasons. The rainy season mainly occurs in summer. Winter has low precipitation and is windy, with the average annual number of gale days being about 70 [52]. Moreover, the solar radiation is intense, and the annual temperature difference is small, while the daily difference is significant ($-27\ °C$ to $31\ °C$). Water resources are quite rich, with a total area of $7.93 \times 10^4\ km^2$. The Brahmaputra River basin has a natural potential hydropower of $2389 \times 10^4\ KW$ [51]. In addition, due to vast terrain fluctuations, precipitation is concentrated and forest land and grassland resources are considerably adequate.

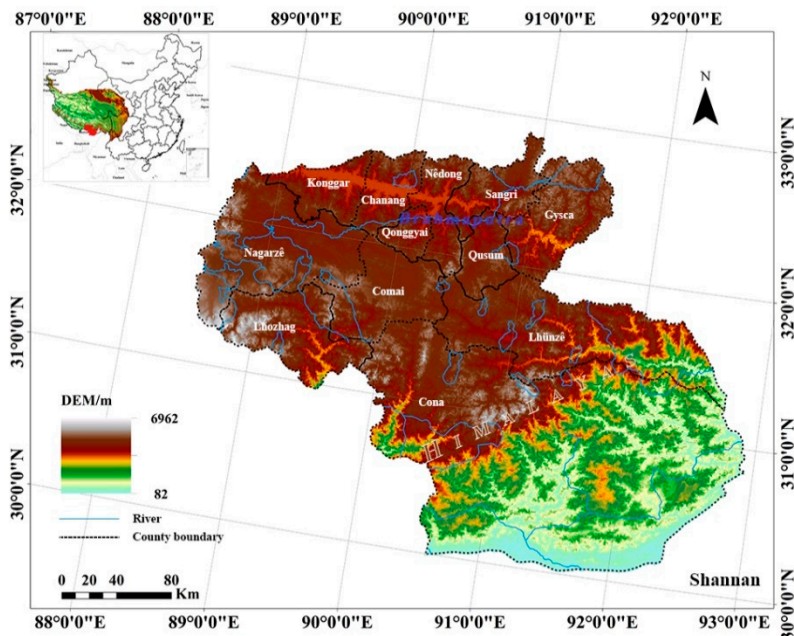

**Figure 1.** Location of study area.

## 2.2. Data Collection

Land-use/land-cover (LULC), population density (POP), digital elevation model (DEM), for 1990, 1995, 2000, 2005, 2010, and 2015, and normalized difference vegetation index (NDVI) data for 2000–2015 were obtained from the Data Center for Resources and Environmental Sciences, Chinese Academy of Sciences, RESDC, Available online: http://www.resdc.cn (accessed on 30 April 2020). The NDVI data for 1990 and 1995 were obtained from the National Tibetan Plateau Data Center of China, TPDC, Available online: http://data.tpdc.ac.cn/ (accessed on 18 May 2020). It is calculated on AVHRR satellite images (the spatial resolution is 1 km). Gross domestic product (GDP) data were obtained from the National Earth System Science Data Sharing Infrastructure, National Science and Technology Infrastructure of China. Meteorological data were provided by the China Meteorological Data Service Center, CMDC, Available online: http://data.cma.cn (accessed on 30 April 2020). Afforestation area data were obtained from the Statistical Year Book of Shannan in 1990–2018. The spatial resolution was $1\ km^2$.

## 2.3. Methodology

### 2.3.1. Assessment and Gradation of Ecological Vulnerability

Given the existing international evaluation principles and standards, the comprehensive evaluation system of ecological vulnerability was established combining the ecological and the social-economic conditions of the study area [16,20,21,24,53,54]. The EVI is based on four aspects with 18 indicators: topography (slope), surface (topographic relief, vegetation coverage, land use degree, landscape diversity, desertification area, plateau permafrost area, grassland area, water resource area), meteorology (average annual precipitation,

relative humidity, average annual temperature, hours of sunshine, wind speed, solar radiation intensity), and human disturbance (population density, GDP, livestock husbandry output) [43,55,56]. The technique flowchart is shown in Figure 2.

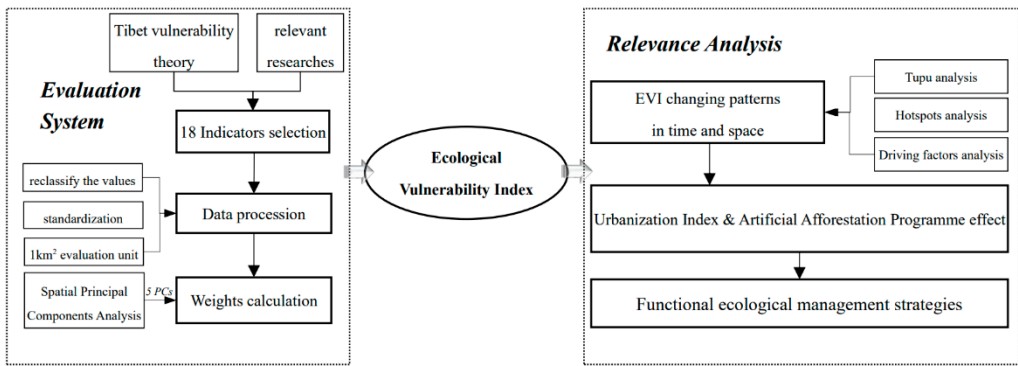

**Figure 2.** The technique flowchart of the study.

The weights of the indicator factors are determined using SPCA (spatial principal component analysis) after the standardization of all indicators in ArcGIS 10.2. SPCA is embedded to determine the weight of each factor and add spatial features on the basis of PCA (principal component analysis). Its calculation principle is consistent with PCA [54,55], which transforms multiple variables into a few principal components through dimensionality reduction and produces a correlation coefficient matrix composed of each standardized index. The first five principal components whose cumulative contribution rate reached 80% or more were selected; the final principal component result is shown in Table 1. The calculation formula was as follows:

$$R = \frac{Z^T Z}{n} \tag{1}$$

$$|R - \lambda I| = 0 \tag{2}$$

$$CCR = \frac{\sum_{j=1}^{m} \lambda_j}{\sum_{j=1}^{n} \lambda_j} \geq 0.80 \tag{3}$$

$$P = Z \times W \tag{4}$$

where $R$ was the correlation coefficient matrix, $Z$ was the standardized value of each selected index, $n$ was the number of indexes, $\lambda$ was the eigenvalues of the $R$ correlation coefficient matrix, $I$ was the identity matrix, $CCR$ was the cumulative contribution rate, $m$ was the number of principal components that were determined, $P$ was the matrix containing values of every considered principal component, and $W$ was m number of eigenvectors with the largest eigenvalues selected to form the matrix.

**Table 1.** The results of spatial principal component analysis.

| Year | Principal Component | 1990 | 1995 | 2000 | 2005 | 2010 | 2015 |
|---|---|---|---|---|---|---|---|
| | I | 0.206 | 0.169 | 0.187 | 0.174 | 0.194 | 0.212 |
| | II | 0.119 | 0.145 | 0.164 | 0.124 | 0.120 | 0.125 |
| Eigenvalue/% | III | 0.083 | 0.069 | 0.067 | 0.059 | 0.061 | 0.055 |
| | IV | 0.045 | 0.048 | 0.049 | 0.054 | 0.053 | 0.050 |
| | V | 0.033 | 0.032 | 0.030 | 0.036 | 0.028 | 0.027 |

**Table 1.** *Cont.*

| Year | Principal Component | 1990 | 1995 | 2000 | 2005 | 2010 | 2015 |
|---|---|---|---|---|---|---|---|
| | I | 36.108 | 30.996 | 32.199 | 32.943 | 35.655 | 37.825 |
| | II | 20.821 | 26.578 | 28.247 | 23.410 | 22.079 | 22.384 |
| Contribution/% | III | 14.491 | 12.695 | 11.446 | 11.120 | 11.183 | 9.812 |
| | IV | 7.987 | 8.735 | 8.421 | 10.297 | 9.752 | 9.029 |
| | V | 5.735 | 5.826 | 5.109 | 6.865 | 5.205 | 4.782 |
| | I | 36.108 | 30.996 | 32.199 | 32.943 | 35.655 | 37.825 |
| Cumulative contribution/% | II | 56.929 | 57.574 | 60.446 | 56.353 | 57.734 | 60.210 |
| | III | 71.419 | 70.269 | 71.893 | 67.473 | 68.917 | 70.021 |
| | IV | 79.407 | 79.004 | 80.314 | 77.769 | 78.669 | 79.050 |
| | V | 85.142 | 84.830 | 85.422 | 84.635 | 83.874 | 83.832 |
| | I | 0.424 | 0.365 | 0.377 | 0.389 | 0.425 | 0.451 |
| | II | 0.245 | 0.313 | 0.331 | 0.277 | 0.263 | 0.267 |
| Weight | III | 0.170 | 0.150 | 0.134 | 0.131 | 0.133 | 0.117 |
| | IV | 0.094 | 0.103 | 0.099 | 0.122 | 0.116 | 0.108 |
| | V | 0.067 | 0.069 | 0.060 | 0.081 | 0.062 | 0.057 |

According to the above method, the higher the *EVI*, the more vulnerable the ecological environment. The index is obtained by the sum of the comprehensive values of multiple principal components and their corresponding weights [57], as shown in the following formula:

$$EVI = \sum_{i=1}^{m} r_m P_m \tag{5}$$

$$r_i = \frac{n_i}{\sum_i^m n_i} \tag{6}$$

where *EVI* is the ecological vulnerability index; *r* is the contribution ratio; *P* is the principal component; *m* is the number of principal components; $r_i$ is the contribution ratio of the *i* principal component; and $n_i$ is the eigenvalue of the *i* principal component.

To quantitatively assess the changing trend of the vulnerability of regional ecosystems, a comprehensive regional ecological vulnerability index (EVSI, Ecological Vulnerability Standard Index) needs to be constructed and standardized according to Equation (7) to obtain the comprehensive ecological vulnerability index EVSI of Shannan City:

$$EVSI_i = \frac{EVI_i - EVI_{min}}{EVI_{max} - EVI_{min}} \tag{7}$$

To facilitate spatial analysis, the *EVSI* was classified into five levels using the natural breakpoint classification method (Jenks) in ArcGIS [58]: slight vulnerability: <0.2786, light vulnerability: 0.2796–0.4418, medium vulnerability: 0.4418–0.6031, high vulnerability: 0.6031–0.76132, and extreme vulnerability: >0.7613.

### 2.3.2. Tupu Analysis of Ecological Vulnerability

Geo-information Tupu analysis is a processing method that combines the spatio-temporal change information of land use through map units to quantitatively express the characteristics of the change [59]. Based on the ArcGIS tool, EVI process change is revealed via Geo-information Tupu. The specific operational formula was:

$$N = F + 10L \tag{8}$$

where *N* represents the newly generated grid Tupu of the ecological vulnerability type change in the research stage; *F* is the attribute value of the ecological vulnerability Tupu grid in the previous period; *L* is the attribute value of the ecological vulnerability Tupu grid in the later period. After the Tupu algebraic superposition operation is fused according to the Tupu unit, the pivot table is obtained, and the transfer matrix is made. At the same

time, the ecological vulnerability type transfer Tupu of Shannan City from 1990 to 2015 is obtained.

### 2.3.3. Cold-Hot Spot Study Change Analysis

Cold-hot spot analysis is a spatial analysis model used to display the degree of spatial aggregation calculated by Getis-Ord Gi*. The results of the EVI calculations are visualized using the cold-hot spot model. In this study, the Getis-Ord Gi* index was used to analyze the high/low spatial aggregation degree of EVI changes, that is, the spatial distribution of cold/hot spots. Moreover, cold-hot spot analysis is the database to underpin the heterogeneity analysis. The calculation formula was:

$$Gi^* = \frac{\sum_{j=1}^{n} w_{ij} x_j - X \sum_{j=1}^{n} w_{ij}}{s \sqrt{\left[ n \sum_{j=1}^{n} w_{ij}^2 - \left( \sum_{j=1}^{n} w_{ij} \right)^2 \right] / (n-1)}} \tag{9}$$

$$X = \frac{1}{n} \sum_{j=1}^{n} x_j \tag{10}$$

$$S = \sqrt{\left( \frac{1}{n} \sum_{j=1}^{n} x_j^2 - X^2 \right)} \tag{11}$$

where $G_i^*$ was the output statistical Z-score, $x_j$ was the EVI change of space unit $j$, and $w_{ij}$ was the spatial weight between adjacent space units $i$ and $j$.

### 2.3.4. Spatial Correlation Analysis between EVSI and Urbanization Level

According to a previous study [60], the composite urbanization index (UI) can be quantified through population urbanization, economic urbanization, and land urbanization. These three levels are expressed via population density (PD), GDP density (GDPD), and area occupied by built-up land (ULP), respectively.

$$UI = \frac{1}{3} \times \left( PD' + GDPD' + ULP' \right) \tag{12}$$

where $UI$ represents the urbanization index of the evaluation unit and $PD'$, $GDPD'$, and $ULP'$ represent the population density, $GDP$ density, and construction land area ratio of the evaluation unit after standardization.

The spatial correlation between EVSI and UI was analyzed using Moran's I index. If Moran's I > 0, it means that there is a positive correlation between them, and vice versa, there is a negative correlation; also, the spatial clustering types of EVSI and UI were obtained in GeoDa software, which was classified as insignificant, high-high, low-low, low-high, and high-low. The formula is as follows:

$$I = \frac{N \sum_{i}^{N} \sum_{j \neq i}^{N} W_{ij} Z_i Z_j}{(N-1) \sum_{i}^{N} \sum_{j \neq i}^{N} W_{ij}} \tag{13}$$

$$I_{kl}^i = Z_k^i \sum_{j=1}^{N} W_{ij} Z_l^j \tag{14}$$

$$Z_k^i = \frac{X_k^i - X_k}{\sigma_k} \tag{15}$$

$$Z_l^j = \frac{X_l^j - X_l}{\sigma_l} \tag{16}$$

where $I$ is the global bivariate Moran's I for the EVSI and urbanization and $I_{kl}^i$ is the local bivariate Moran's I for the EVSI and urbanization level. $N$ stands for the total number of spatial units. $W_{ij}$ stands for spatial weight matrix for measuring spatial correlation between the $i$ and $j$ spatial unit. $Z_i$ refers to the deviation between the attribute of $i$ spatial unit and the average of the attribute. $Z_j$ refers to the deviation between the attribute of $j$ spatial unit and the average of the attribute. $X_k^i$ refers to the value of attribute k of spatial unit $i$; $X_k$ refers to the average of attribute $k$; $\sigma_k$ is the variance of attribute k. $X_l^j$ refers to the value of attribute $l$ of spatial unit $j$; $X_l$ refers to the average of attribute $l$; $\sigma_l$ refers to the variance of attribute $l$.

## 3. Results

### 3.1. Spatial and Temporal Changes of Ecological Vulnerability

The EVI for Shannan City reached its highest level in 2000 and lowest in 1990, and overall, Shannan City was at a medium vulnerability level during the study period.

The percentages of areas with different types of ecological vulnerability for 1990–2015 are shown in Figure 3. On the temporal scale, in 1990 and during the period 2005–2015, the largest percentage of areas had light vulnerability (slight-vulnerability areas being the lowest, except in 2000). Among five ecological vulnerability types, light vulnerability had the lowest area fluctuation, with a decrease of 0.13% (high vulnerability and extreme vulnerability had the biggest area fluctuation); in 1995, the largest percentage of areas were in medium vulnerability, and by 2000, the high-vulnerability areas occupied the largest area.

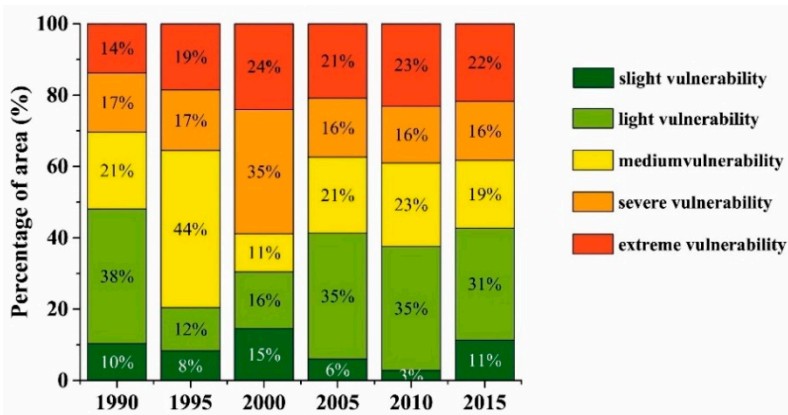

**Figure 3.** Proportion of ecological vulnerability of Shannan in 1990–2015.

Figure 4 shows the spatial variation of ecological vulnerability. During the study period, the EVI showed a significant downward trend in most of the northwest and northern regions of Shannan City, and the high-vulnerability areas migrated southward. In 1990–2000, the EVI mainly showed a west-high and an east-low state, while in 2005–2015, it mainly showed an increasing spatial pattern along the northwest–southeast axis.

Spatial distribution of different EVI types showed that slight-vulnerability areas were mainly located in river valleys in central and northwestern regions of Cona County at lower altitudes in 1990–2000 and in river valleys in northwestern Shannan City in 2005–2015. Light-vulnerability areas were mainly concentrated at low altitudes and in water bodies in central Cona County, while medium-vulnerability areas were mainly found in the southern part of the Himalayas, where there is a large elevation difference across the mountain range. High-vulnerability areas were mainly located in Nagarze County, Konggar County, northern Chanang County, northwestern Comai County, and Lhunze County, while extreme-vulnerability areas were concentrated in Nagarze County and Lhunze County in 1990–2000 and in the southeastern margin and northern part of the Himalayas in 2005–2015.

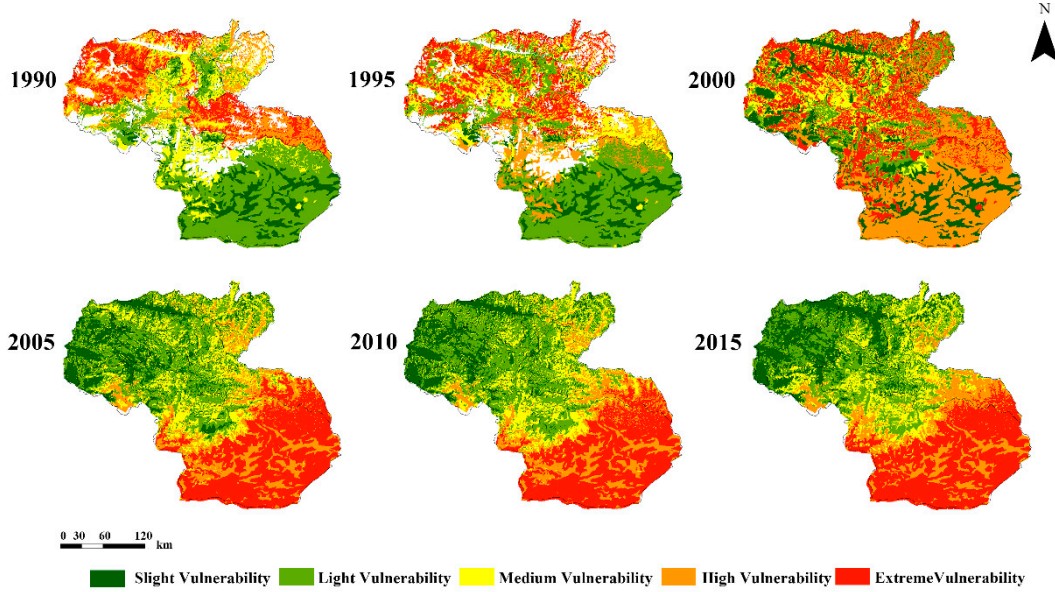

**Figure 4.** The EVI distribution of Shannan in 1990–2015.

### 3.2. Transformation of EVI

The Tupu model of the ecological vulnerability level dynamics in Shannan City from 1990 to 2015 is shown in Figure 5. On the temporal scale, from 1990 to 2015, the ecological vulnerability Tupu transformed areas in 77.67% of the study region, with 20 types of shifts, the predominant being the shift from slight to extreme vulnerability, accounting for 25.86% of the total transformed area. This change was followed by a shift from extreme to light vulnerability, accounting for 10.19% of the total transformed area. Every five years, the transformed areas of ecological vulnerability types accounted for 73.27%, 57.64%, 78.57%, 59.50%, and 69.89% of the study area, respectively, with the change from slight to medium vulnerability being the primary type in 1990–1995 and 1995–2000. From 2005 to 2010, there was a decrease in the number of transformed areas at the EVI level, with the primary shift being from light to slight vulnerability in the study area.

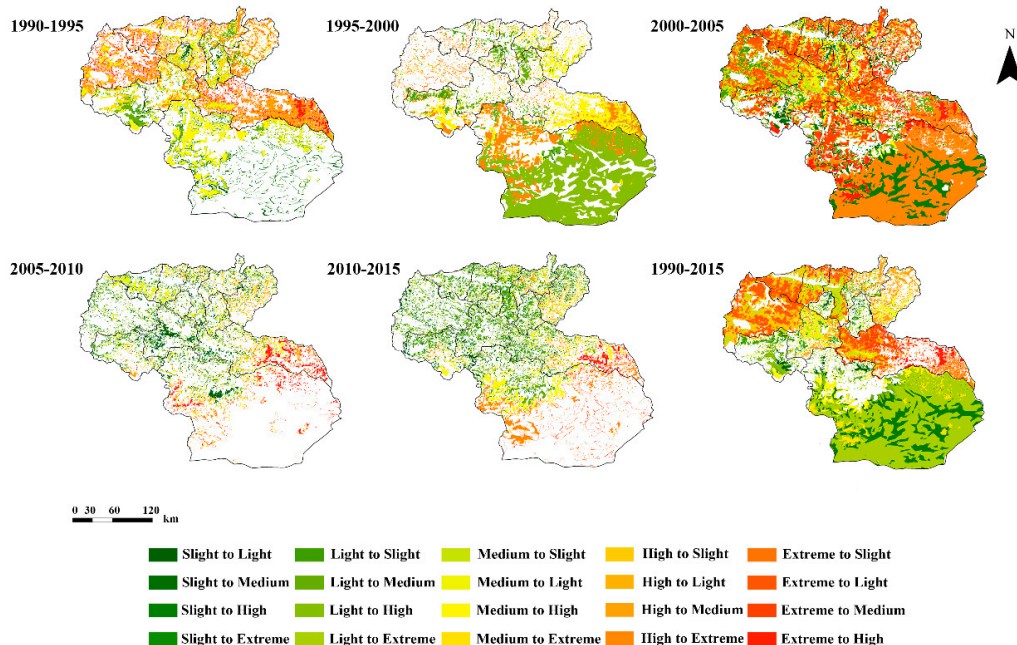

**Figure 5.** EVI Tupu changes of Shannan in 1990–2015.

On the spatial scale, Tupu changes in the EVI in 1990–2015 in Shannan City were characterized and more concentrated, with the northern and central regions improving overall, while southern Cona County became severely ecological vulnerable, especially in 1995–2000. Overall, the EVI showed the largest indications of improvement as the largest proportion of extreme to slight vulnerability transformation in Cona County and northern Shannan City. In 2005–2015, the EVI was comparatively stable, with transformed areas scattered mostly in the middle of Lhunze County and Comai County. In 1995, changes in Tupu showed that the previous and subsequent stages of 1995 were significantly different. Namely, in 1990–1995, the transformed areas mainly clustered in northern Shannan City, with an improvement trend, while in 1995–2000, northern Shannan City had few transformed areas, while southern Shannan City witnessed a significant decrease in the EVI.

### 3.3. Spatial Heterogeneity Analysis of Ecological Vulnerability

The distribution of cold/hot spots of ecological vulnerability in Shannan City during the study period is shown in Figure 6. Overall, the hot-spot center moved from north to south, and the cold spots clustered in the northern region of the study area. Temporal changes demonstrated that the hot spots showed a decreasing–increasing–decreasing trend from 1990 to 2015, with the largest area of hot spots reaching its peak in 2005. Meanwhile, the area with a comparatively high ecological vulnerability index increased over the 25 years of the study period. The cold spots showed an increasing trend from 1990 to 2015, with the largest area of cold spots in 2015, indicating an overall increase in the ecological vulnerability index in Shannan City during the study period.

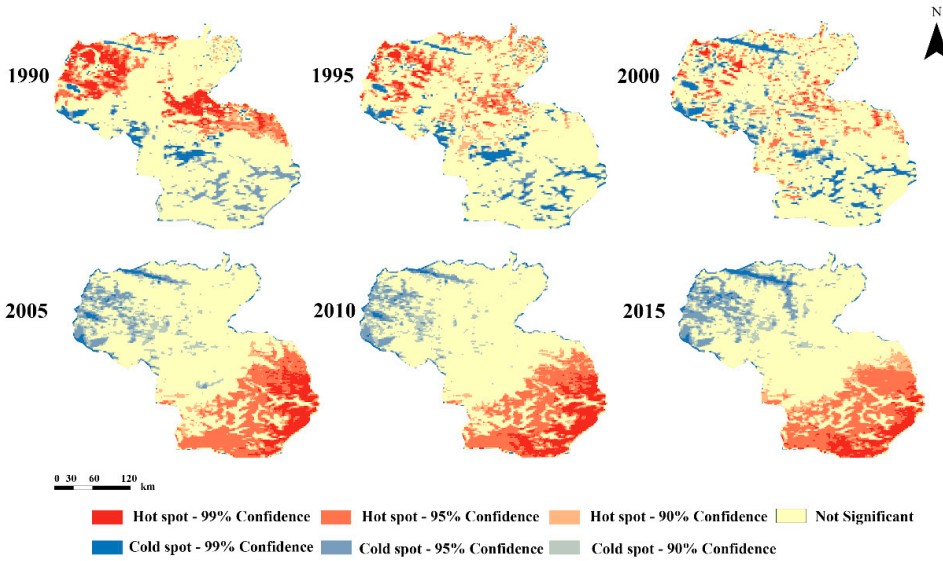

**Figure 6.** Analysis of cold-hot spots of EVI changes in Shannan from 1990 to 2015.

Spatial changes showed that the spatial distribution and area proportion of cold spots of the EVI of Shannan City changed significantly around 2000. Namely, in 1990–2000, the hot spots of the EVI were mainly concentrated in northern and eastern Lhunze County, and there was a significant increase in the EVI of Shannan City, while cold spots were mainly distributed in south and central-northwest Shannan City. Although there was a gradually decrease in significant EVI change areas in northern high-altitude areas with low vegetation cover, the overall EVI still increased due to a significant increase in the EVI in some of the areas that underwent severe changes and became highly vulnerable. In particular, the EVI in southern Shannan City (Cona County) evidently increased from 2005, which needs more attention and solution strategies. In contrast, cold spots were mainly located in the northwestern region (Comai County and Nagarze County), with an

increasing trend. This pattern indicates that the EVI in northwestern and eastern Shannon City improved distinctly and gradually during the study period.

### 3.4. Determinant Factors of EVI

The principal component analysis results are shown in Figure 7. The first five principal component layers, which accumulated a contribution of over 80%, were used to explore the drivers of change in ecological vulnerability in Shannan City.

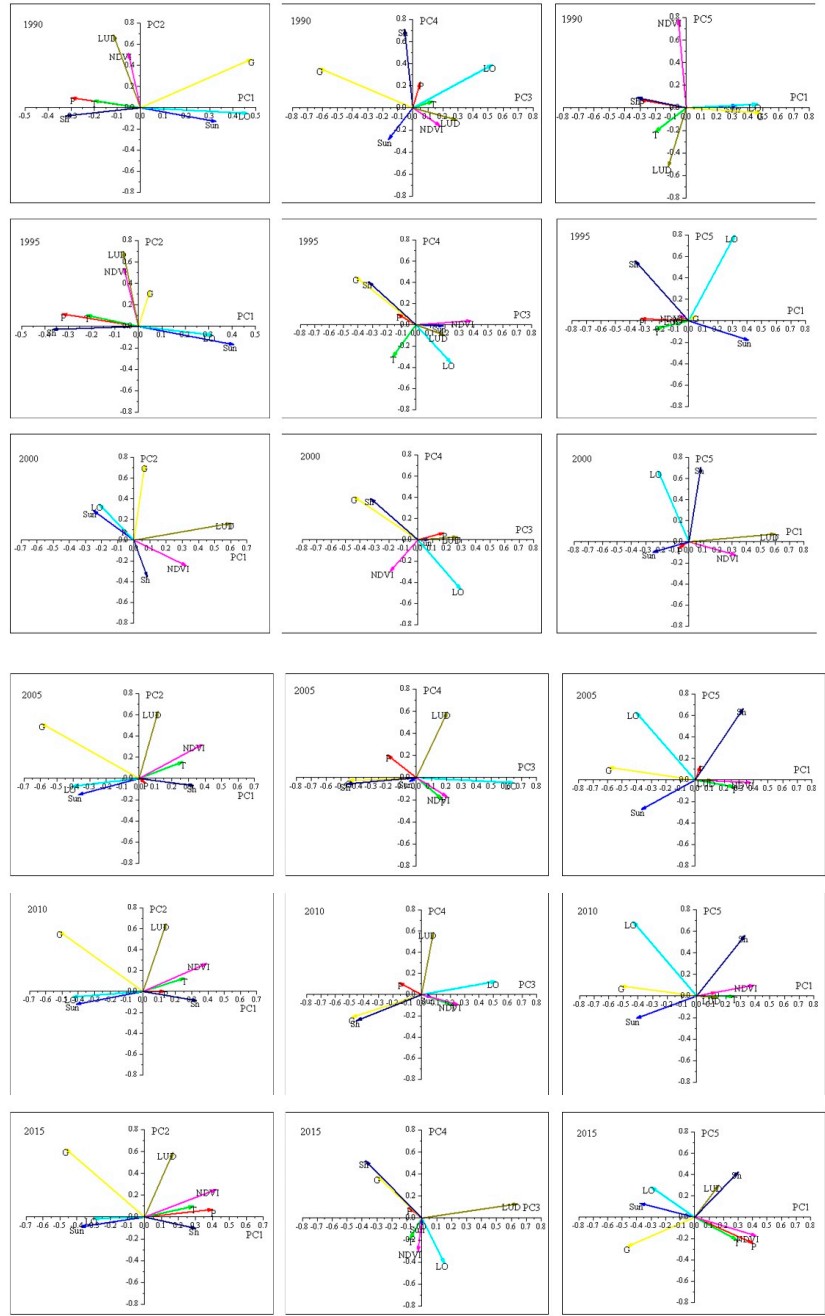

**Figure 7.** Correlation analysis between principal components 1–5 and various indicators. (P—average annual precipitation; T—average annual temperature; Sun—hours of sunshine; POP—population density; LO—livestock husbandry output; S—slope; NDVI; GDP; G—grassland area; PPA—plateau permafrost area; W—wind speed; SRD—surface relief degree; SRI—solar radiation intensity; LUD—land use degree; DA—desertification area; Sh—Shannon index; RH—relative humidity; WRA—water resources amount).

In principal component 1 (PC1), the EVI was negatively correlated with the land-use degree (−0.1171) and the NDVI (−0.0518) in 1990–1995, and grassland area (0.4813) and livestock husbandry output (0.4614) factors played the most positive role. In 2000 and after, the positive correlation between the EVI and land-use degree became more vital, while the correlation with the EVI was the opposite for the livestock husbandry output factor.

PC2, PC3, and PC4 of the EVI were always positively correlated with the land-use degree. PC3 and PC5 of the EVI were always positively correlated with livestock husbandry output. PC1, PC2, and PC4 of the EVI were positively correlated with grassland area from 1990 to 2000, but PC3 and PC5 were negatively correlated. This trend changed in a diametrically opposite direction in the period 2005–2015. Among 18 determinants since 1995 in PC3 and PC4, the desertification area factor became critical and reached a peak in 2005 and 2010 in PC4 at 0.7216 and 0.7001, respectively. In addition, the Shannon City index factor was vital throughout the study period in PC4 and PC5, with correlation indexes of 0.7114, 0.5582, 0.7046, 0.6642, 0.5618, and 0.5187.

Analysis of the integrated correlation index of PC1–PC5 in 1990–2015 showed that the most influential factor of the EVI is the land-use degree, having the largest correlation indexes of 0.3083, 0.2882, and 0.3651 in 2005, 2010, and 2015, respectively. In 1990, 1995, and 2000, the most vital determinant factors of the EVI were livestock husbandry output, desertification area, and average annual temperature, with the correlation indexes of 0.2718, 0.1968, and 0.2278, respectively.

The results showed that livestock husbandry output, land-use degree, and desertification area have been prominent since 1995. The grassland area and land-use degree contribute significantly to each of the principal components every year as well. Therefore, based on these results, land-use degree, livestock husbandry output, desertification area, and grassland area are the main factors influencing the EVI.

*3.5. Changes of NDVI and Afforestation Area*

The NDVI reflects the absorption and reflection characteristics of vegetation in the red and near-infrared regions and, therefore, provides a good indication of ground vegetation growth [61–63]. The annual and seasonal NDVI averages reflect vegetation growth during the year and season [64]. The NDVI of forest land can reflect the growth status and spatial distribution density of forest trees, etc. The NDVI can also reflect the effectiveness of the AAP policy. The time series is considered a linear regression function of time: **Y = at + b**. The interannual trend line equation of the NDVI is shown in Table 2. The changes in the EVI and afforestation area from 1990 to 2015 are shown in Figure 8.

**Table 2.** The NDVI and afforestation area inter-annual trend line equation.

| Index | Formula |
| --- | --- |
| NDVI | y = 0.0032t − 5.8989 |
| Afforestation area | y = 0.0351t − 69.994 |

t stands for the year; where a is the tendency value, a > 0 indicates an increasing trend with time; a < 0 indicates a decreasing trend with time, and the absolute value of a value reflects the rate of increase or decrease.)

Table 2 and Figure 8 can tell that the NDVI in Shannan during the 25 years has a positive trend in all, and the afforestation area has an obvious positive trend as well. This means that there is a better turn in the vegetation growth status in afforested areas, and the spatial distribution density also showed an increasing trend in the last 25 years. What's more, the negative relevance between the afforestation area and the EVI is shown in Figure 8, which indicates that when the afforestation area increased, the EVI decreased at the 5 years.

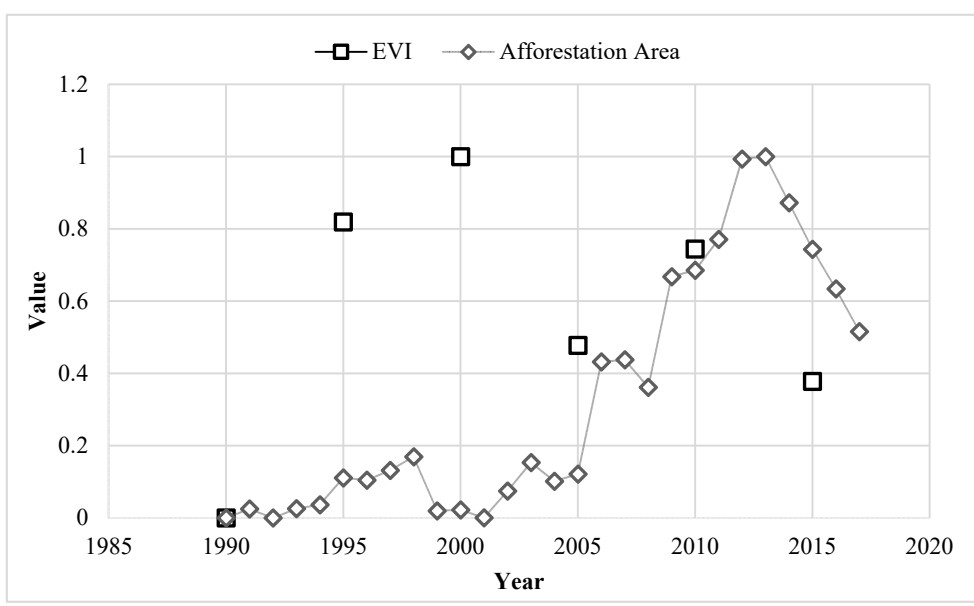

**Figure 8.** The changes of EVI and afforestation area from 1990 to 2015.

### 3.6. Impact of Urbanization on Ecological Vulnerability

From 1990 to 2015, both the EVI and the urbanization level in Shannan City increased significantly, indicating that both show a synergistic change over timescales. However, the pattern of the spatial response of the EVI to urbanization is unknown. The EVI and the urbanization level were positively correlated in 1990–1995 but negatively correlated in 1995–2015, and the negative correlation gradually strengthened. The Z-values in Table 3 show that the EVI and urbanization level had a powerful spatial aggregation effect, with the weakest only in 2000.

**Table 3.** Moran's I index and significance test results of Shannan City's EVI and urbanization levels from 1990 to 2015.

| Index | Year | 1990 | 1995 | 2000 | 2005 | 2010 | 2015 |
|---|---|---|---|---|---|---|---|
| | Moran's I | 0.604 | 0.213 | −0.007 | −0.340 | −0.316 | −0.379 |
| EVI&Urbanization | \|z-Value\| | 188.7587 | 76.4382 | 2.8697 | 128.995 | 121.7327 | 139.3639 |
| | *p*-Value | 0.001 | 0.001 | 0.002 | 0.001 | 0.001 | 0.001 |

*p*-values indicate the probability of an event occurring, $p < 0.05$ indicates a statistically significant difference, and $p < 0.01$ indicates a statistically significant difference. z-values are multiples of the standard deviation reflecting data dispersion or aggregation, $|Z| > 2.58$, which corresponds to $p < 0.01$, indicating a statistically significant difference.).

## 4. Discussion

### 4.1. The Spatial-Temporal Patterns of Ecological Vulnerability

This study identified the spatial and temporal dynamic patterns of ecological vulnerability in Shannan City and confirmed the significant temporal and spatial heterogeneity of the EVI in the region.

In terms of the time series, the variation in different EVI types changing over time was significant (Figures 3 and 4). This condition may result from differences in human disturbance (or natural effects) over the years, with the land-use degree and livestock husbandry output being the main factors affecting the EVI. That is, from 1990 to 2000, unsustainable, large-scale urbanization infrastructure construction and mining were launched in northern Shannan City to meet the needs of urbanization [65]. Ecological lands (e.g., grasslands, wetlands) were converted to construction lands, decreasing light-vulnerability areas. However, from 2000 to 2015, when Shannan City started implementing ecological conservation policies [66], the forest cover increased, transitioning extreme-



vulnerability areas in northern Shannan City to slight- and light-vulnerability areas in 2005. In addition, the slight- and light-vulnerability areas are mainly in and around water bodies, so it is essential to protect water resources. Therefore, there was a decreasing trend in the EVI from 1990 to 2000 but a reversal in the trend after 2000 in the northern study region.

However, the EVI's heterogeneity is more pronounced spatially, with extreme-vulnerability areas clustered mainly in the north, with a concentrated population and urban areas. Similarly, medium-vulnerability areas are generally located in central Shannan City, with relatively inaccessible regions (e.g., protected areas). Light-vulnerability areas, which account for the largest proportion of the area, decrease from east to west. In addition, slight-vulnerability areas, which account for the smallest proportion of the area, mainly include stable valleys, open water resources, and zones with low landscape diversity and predominantly grassland use (Figure 4). These findings are consistent with previous studies. Li et al. stated that human activities affect local ecosystems by transforming natural ecosystems into those that are artificial or semi-artificial [67]. In light of Li et al.'s findings, the areas with the most significant spatial variation of the EVI of Shannan City are mainly those where the Brahmaputra River runs through (along southern Nagarze County to eastern Gyaca County) and those with high population density in the north (Nagarze County and Nedong County). In these areas, the overall geomorphological pattern is a wide alpine valley, where ditches and the main river meet to form flooded fans and broad mountain valleys, with intense sandstorm activity, loose soil texture, and high ecological vulnerability. Therefore, with heavy rainfall in the rainy season, the riverbank erodes, and heavy and continuous rainfall leads to landslides and other geological disasters. This heavy rainfall also causes severe desertification, aggravating these areas' EVI to some extent.

### 4.2. Probable Driving Factors of Ecological Vulnerability

Analysis of the EVI's driving factors showed that the EVI is significantly and positively correlated to the land-use degree, livestock husbandry output, and desertification area. However, the impact of urbanization on the EVI of Shannan City showed that the EVI was not significantly affected by the urbanization level over the 25 years of the study period (positive in 1990–1995 and negative in 2000–2015), even when the land-use degree was the most significant impact factor for the EVI. The positive correlation of the urbanization level with the EVI in the early period (1990–1995) caused some unchangeable residential area patterns, which further changed the land-use patterns in subsequent years.

How did the land-use degree and land-use patterns gradually impact the EVI of Shannan City for 25 years and become the most significant determinants of the EVI? In the statistical yearbook, in 1990–1995, although the dominant production pattern in Shannan City relied heavily on pastoral industries [68], the land-use degree as the most critical factor that affects the EVI changed mostly for urbanization. In addition, the construction land scale expanded too quickly [65]. Due to the penetration of the Himalayas, the elevation difference of Shannan City is significant. The lowest elevation of Cona County in the south of the Himalayas is 82 m, and the highest peak is 6962 m. This topography makes the distribution of residential areas in Shannan City quite obvious. Namely, villages and towns dominate the northern alpine region. In addition, due to the Brahmaputra River's passage and the remarkable altitude difference, northern Shannan City, as an ideal construction site for hydropower projects, has more water conservancy projects [69]. Therefore, the residents' electricity use is guaranteed, and the urbanization level is relatively high. Hydropower construction projects have also intensified soil erosion and led to the desertification of both riverbanks [70], which is in line with the positive correlation between urbanization level and EVI.

However, in the south, with high vegetation cover, abundant rainfall, and warm climate, livestock husbandry and agriculture are developed and residents are mainly nomadic herders. Livestock husbandry output and a series of factors related to pastoral area changes, such as grassland area, desertification area, and land-use degree, are the main factors affecting the EVI. Consequently, the urbanization level in this area has relatively little correlation

with the EVI, showing a negative correlation as a whole. The southern Cona County border has not yet been determined, so Indians often live and graze in southern Cona County [71]. Overgrazing causes drastic changes and a severe decrease in southern Shannan City's grassland area, significantly increasing ecological vulnerability [72]. Further, the AAP has reduced the EVI of Shannan City by enhancing the vegetation cover (afforestation area). Over the 25 years of the study period, the biggest growth of the EVI occurred in 2000, with a rate of 122.1%, which was also the year with the largest drop in the afforestation area (79.9%). In general, the EVI of Shannan City has remained in a sub-vulnerable state, showing an improving trend, while the afforestation area demonstrated an obvious increase of 393.9%. This indicates that, with the launch of the AAP, EVI changes and areas of forests became directly proportional. This correlation strengthened in 25 years, with the most obvious bond in 2000, emphasizing the effectiveness of the AAP.

In summary, geological disasters, such as overgrazing and land-use changes, such as urban construction, have increased the EVI of Shannan City, further causing a polarization of Shannan City's EVI distribution, shifting the EVI center from north to south. However, these factors have been weakened to some extent by implementing the AAP. Meanwhile, soil erosion due to sandstorms and overgrazing and extreme altitude differences are common in the Qinghai–Tibet Plateau as well. These factors also contribute to the severe ecological vulnerability of other regions of the Qinghai-Tibet Plateau.

*4.3. Sustainable Implications for Ecosystems Management*

As the Qinghai–Tibet Plateau border area, Shannan City is penetrated by the Himalayas and is located on the national border. Its extreme climate and altitude make Shannan City highly susceptible to ecological vulnerability. The study results indicate that the EVI of Shannan City is mainly affected by the land-use degree, grassland area, livestock husbandry output, and desertification area. These factors are related to irrational land-use and planning management, such as unregulated grazing, massive construction, and undeveloped, fragmented industrial parks [65]. If not planned reasonably, the land-use waste situation in Shannan City will become increasingly severe, and the EVI will increase. Therefore, combined with the current goals and needs of promoting coordinated and sustainable development of the economy, society, resources, and the environment, this article gives the following suggestions to prevent the vulnerability of the ecological environment of Shannan City from increasing further.

First, from the perspective of improving the land-use degree and more adequately developing land resources, measures should focus on sustainable land intensification and pastoral management, in addition to managing high- and extreme-vulnerability zones, strengthening supervision of the medium-vulnerability zone, and focusing on defensive measures in light- and slight-vulnerability zones. To do so, the land spatial planning structure should be modified. The land in cities should be rezoned by the principle of different functions into industrial, commercial, residential, and tourist areas. In towns and villages, it is crucial to control the property of premises, eliminate illegal occupation of cultivated land or even permanent bare farmland, and expand the cultivated land area, making cultivated land part of the income source of nomads. Professional and technical personnel should be employed to support scientific and technological innovation and conduct ecosystem vulnerability evaluation and research. In addition, the government should perform appropriate related planning to promote the ecological restoration of land and space and accelerate the construction of a spatial planning system to strengthen environmental protection and rational planning and development of land resources. The government should also build artificial meadows and protect cropland to enable nomads to transform or combine as much as possible to settle and semi-settle. This way, the land-use cost can be determined and the land-use degree improved.

Second, from the perspective of desertification caused by humans and geological disasters, the following can be adopted:

(1) Afforestation construction. Conserving water sources and reducing the risk of geological disasters can significantly alleviate the ecological vulnerability of Shannan City.

(2) Inspections according to period and area, early warning and forecasts of meteorological and geological disasters, and strict implementation of the assessment of geological disasters of construction projects should be carried out. Measures should be taken to manage and prevent sudden geological disasters in a short period of time.

(3) The education and training of livestock husbandry producers and organizers should be strengthened for them to fully understand the importance and urgency of preventing grassland degradation and maintaining grassland ecological functions.

Therefore, measures such as rational environmental planning, building and improving forestry and grassland systems, and strengthening water resources are critical for controlling the increase in ecological vulnerability, promoting sustainable development of green cycles, and forming an industrial structure and a stable and sound ecosystem environment. In addition, these measures can alter the ecological vulnerability of Shannan City into a beneficial cycle.

## 5. Conclusions

Ecological vulnerability varies greatly both temporally and spatially with changes in the natural landscape configuration and anthropogenic disturbances such as grazing. This study analyzed the temporal and spatial variation, heterogeneity, and driving factors of the EVI through a new, integrated evaluation system to quantify ecological vulnerability. The analysis leads to the following conclusions:

(1) Shannan City's ecological vulnerability level did not significantly change in the time series and generally remained at medium vulnerability, with the EVI reaching its highest value in 2000 and lowest in 1990. The EVI is more embodied spatially than temporally. The spatial distribution of the EVI from 1990 to 2000 mainly manifested as a higher-vulnerability state in the west than that in the east. From 2005 to 2015, it mainly manifested as an increasing trend along the northwest–southeast axis.

(2) During the study period, 77.67% of Shannan City had been transformed with 20 types of changes, the dominant type being the shift from slight to extreme vulnerability (25.86% of the total transformed area) concentrated in Cona County. The northern and central regions improved overall, while southern Cona County became severely ecologically vulnerable.

(3) The hot-spot center moved from north to south, and cold spots clustered in the northern part of the study area in 1990–2015. The hot spots reached the largest area in 2005 and the cold spots in 2015, indicating gradual stabilization of the EVI after 2005.

(4) The EVI determinant factors for Shannan City are significantly correlated with the land-use degree, livestock husbandry output, grassland area, and desertification area and are weakly correlated with the urbanization level. The Artificial Afforestation Program has a positive effect by preventing the environment from becoming more vulnerable.

This study provides a scientific basis for resource development, ecological environment protection, and planning and construction. Therefore, this study can provide policy application recommendations for coordinating ecosystem protection and socio-economic development, helping Shannan City achieve a sustainable state and improve its ecological vulnerability to benefit the Qinghai–Tibet Plateau's relevant areas.

**Author Contributions:** Conceptualization, Yunxiao Jiang and Luo Guo; Formal analysis, Yu Shi and Luo Guo; Funding acquisition, Luo Guo; Investigation, Yunxiao Jiang; Methodology, Yunxiao Jiang and Rong Li; Project administration, Luo Guo; Resources, Yunxiao Jiang, Yu Shi and Luo Guo; Writing–original draft, Yunxiao Jiang; Writing–review & editing, Rong Li and Luo Guo. Conceptualization, Yunxiao Jiang and Luo Guo; formal analysis, Yu Shi and Luo Guo; funding acquisition, Luo Guo; investigation, Yunxiao Jiang; methodology, Yunxiao Jiang and Rong Li; project administration, Luo Guo; resources, Yunxiao Jiang, Yu Shi and Luo Guo; writing—original draft, Yunxiao Jiang; writing—review and editing, Rong Li and Luo Guo All authors have read and agreed to the published version of the manuscript.

**Funding:** This research study was supported by the Second Tibetan Plateau Scientific Expedition and Research (STEP) program (2019QZKK0308), and innovation team project of the Chinese Nationalities Affairs Commission, grant number 10301-0190040129.

**Institutional Review Board Statement:** Not applicable.

**Informed Consent Statement:** Not applicable.

**Acknowledgments:** We are grateful for the comments of the anonymous reviewers, which greatly improved the quality of this paper.

**Conflicts of Interest:** No conflict of interest exists in this manuscript, and the manuscript is approved by all authors for publication. The authors declare that the work described was original research that has not been published previously and is not under consideration for publication elsewhere, in whole or in part.

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
