# Peer review of "Natural and Political Determinants of Ecological Vulnerability in the Qinghai–Tibet Plateau: A Case Study of Shannan, China"

_ijgi, doi:10.3390/ijgi10050327_

Round 1

Reviewer 1 Report

Dear authors,
I hope you're well
I enjoyed reading your article and appreciate your scientific effort to produce this manuscript. The article language is well structured and well written. I only recommend the following suggestions to improve the research:

1. It is not recommended to use a pronoun (i.e. we and our) when writing research papers. Formal writing is almost always written in the third person.

2. I recommend you put the same colours in the legends of Figure 2 and Figure 3. It is easier to interpret.

3. You will find some new related references which can be added to the literature review: Santos, R. M. B., Sanches Fernandes, L. F., Vitor Cortes, R. M., & Leal Pacheco, F. A. (2019). Hydrologic impacts of land use changes in the Sabor River Basin: A historical view and future perspectives. Water11(7), 1464. https://doi.org/10.3390/w11071464

Author Response

Dear Editor,

Thank you very much for your time and comment on our manuscript. I am submitting the revised manuscript along with the response file. Below are the point-by-point responses to the comments. The manuscript was proofread with assistance from MDPI English editing.  

Reviewer 2 Report

p.2

Line 50 “1 km2” → why “1 km2”?

Line 54-55 “The Closing Hill for Afforestation program and the artificial afforestation program (AAP) play an increasingly important role as important means of land greening” → more references are needed.

Line 56 “are also some of China’s largest and best-invested environmental projects [25].” → more updated references are needed.

p.3

Line 124-126 “Given the existing international evaluation principles and standards, the comprehensive evaluation system of ecological vulnerability was established combining the ecological conditions of the study area” → relevant references should be reviewed and elucidated.

Line 126-129 “The EVI is based on four aspects with 18 indicators: Topography (Slope), Surface (Topographic relief, Vegetation coverage, Land use degree,  Landscape diversity, Desertification area, Plateau permafrost area, Grassland area, Water resource area), Meteorology (Average annual precipitation, Relative humidity, Average annual temperature, Hours of sunshine, Wind speed, Solar radiation intensity), and Human disturbance (Population density, GDP, Livestock husbandry output)” → why these four aspects? 18 indicators? Relevant references should be cited and the methodology should be explained.

p.4

“Table 1: what does “1990a, 1995a…” mean?

p.5

Line 174-175 “…the transfer type Tupu of the ecological vulnerability Tupu of Shannan City…” → meaning unclear.

p.7

Line 228 “2000-2015” → “2005-2015”

p.10

Line 329 “…the main indicator factors…” → “…the main factors…”

p.12

Line 337-339 “The NDVI reflects the absorption and reflection characteristics of vegetation in the red and near infrared regions and, therefore, provides a good indication of ground vegetation growth [51]” → more updated references should be cited.

p.14-15

Line 469-471 “These factors are related to irrational land-use and planning management, such as unregulated grazing, massive construction, and undeveloped, fragmented industrial parks” → references should be cited.

p.15

Line 506 “rational environment planning…” → “rational environmental planning”

p.17

Line 602 “Analysis of the contribution of artificial afforestation to forest cover” → “Analysis on Contribution of Artificial Afforestation to Forest Coverage” (as the author’s translation)

Line 604 “Xiaobin Li, Chen Zhang, Catena, https://doi.org/10.1016/j.catena.2020.105066” → “Xiaobin Li, Chen Zhang, Effect of natural and artificial afforestation reclamation on soil properties and vegetation in coastal saline silt soils. Catena, https://doi.org/10.1016/j.catena.2020.105066”

Author Response

(The authors gave the same response as above.)

Reviewer 3 Report

The manuscript focuses on monitoring and evaluating the ecological vulnerability index in a selected locality. Due to the society-wide discussion, the topic of contributions is topical.
Suggestions and comments:
1. Row 31 - it would be appropriate to unify the citation system - [] or Bryan et al. (2001)
2. It may be appropriate to illustrate the procedure and methods used through the flowchart (input data, output) for a better explanation.
3. Were the input data as NDVI taken or were they calculated on the basis of satellite images?
4. Due to the fact that the analysis covers a longer period of time (1990 - 2015), there were no signs or observable values of possible climate change in the studied area?

Author Response

(The authors gave the same response as above.)
